# Phenotypic and genotypic analysis of drug resistance in *M. tuberculosis* isolates in Gansu, China

Yousheng Peng[1], Chenchen Li[1], Xueke Hui[2], Xiaoning Huo[3], Nigus Abebe Shumuyed[4], Zhong Jia[1,5]*

1 Gansu Agricultural University, Lanzhou, Gansu, China, 2 Lanzhou Maternal and Child Health Care Hospital, Lanzhou, Gansu, China, 3 The Third People's Hospital of Lanzhou City, Lanzhou, Gansu, China, 4 Lanzhou Veterinary Research Institute, Chinese Academy of Agricultural Sciences, Lanzhou, Gansu, China, 5 The Second People's Hospital of Lanzhou City, Lanzhou, Gansu, China

* lz-jiaz@163.com

**Data Availability Statement:** All relevant data are within the paper and its Supporting Information files. Additional large data sets are made available via the Dryad public repository, DOI:10.5061/dryad.r4xgxd2np.

## Abstract

Tuberculosis has posed a serious threat to human health. It is imperative to investigate the geographic prevalence of tuberculosis and medication resistance, as this information is essential for informing strategies for its prevention and treatment. Drug resistance was identified using a proportion method. Drug-resistant genes and pathways were predicted using whole genome sequencing. The drug resistance range of bedaquiline was identified using the microporous plate two-fold dilution method, and drug resistance genes were studied using sequencing. The study revealed that 19.99% of the tuberculosis cases had multidrug resistance. The genes of *M. tuberculosis* are predominantly involved in the synthesis of ABC transporters, two-component systems, and bacterial secretion systems, as well as in energy production and conversion, and lipid transport and metabolism. The genes encode for 82.45% of carbohydrate-related enzymes such as glycoside hydrolases, glycosyl transferases, and carbohydrate esterases. The minimum inhibitory concentration (MIC) of bedaquiline against clinical strains was approximately 0.06 µg/mL, with identified mutations in drug-resistant genes Rv0678, atpE, and pepQ, specifically V152A, P62A, and T222N, respectively. The multidrug resistance tuberculosis development was attributed to the strong medication resistance exhibited. It was concluded that tuberculosis had presented a high level of drug resistance. Phenotypic resistance was related to genes, existing potential genetic resistance in *M. tuberculosis*. Bedaquiline was found to possess effective antibacterial properties against *M. tuberculosis*.

## Introduction

Tuberculosis, one of the most ancient diseases to afflict humans, necessitates a prolonged course of multidrug therapy even in cases of latent infection, despite the extensive history of treatment experience [1]. Estimates suggest that globally, approximately 1.7 billion individuals were infected with *Mycobacterium tuberculosis* (*M. tuberculosis*), leading to over 10 million

**Funding:** This study was supported by Natural Science Foundation of Gansu Province (18JR3RA417). The funders had no role in study design, data collection and analysis, decision to publish, or preparation of the manuscript.

**Competing interests:** The authors declare that there are no conflicts of interest regarding the publication of this paper.

new cases of the disease annually and resulting in latent tuberculosis in 2.2 billion individuals [2]. Drug-resistant tuberculosis poses a significant public health challenge in numerous countries, with the increasing prevalence of drug-resistant *M. tuberculosis* complicating the treatment of tuberculosis [3]. The treatment effect of multidrug resistance tuberculosis is poor and the treatment cycle is long [4]. However, a small number of patients with multidrug resistance to tuberculosis were treated with medications such as bedaquiline, linezolid, and clofaczimine, among others, resulting in enhanced therapeutic outcomes as determined by genetic and phenotypic drug susceptibility testing [5]. Drug-resistant tuberculosis responds well to bedaquiline, an anti-tuberculosis medication [6]. While bedaquiline is currently included in the treatment regimen for tuberculosis, its clinical utilization remains restricted due to limited experience [7–9]. It specifically interferes with the action of F-ATP synthase, which prevents the synthesis of ATP [10,11]. Bedaquiline demonstrates limited cross-resistance with other anti-tuberculosis medications in vitro due to its specific enzyme target. Furthermore, bedaquiline has been shown to enhance the therapeutic effectiveness of linezolid, thus highlighting its potential utility in the treatment of tuberculosis and reducing the prevalence of drug-resistant strains and associated mortality rates [12,13]. However, phenotypic susceptibility testing and drug resistance gene analysis must be ongoing when new medications are added to the treatment regimen [14]. Mutations in the genes Rv0678 or atpE [15] are the main causes of *M. tuberculosis's* resistance to bedaquiline. However, mutations in the gene PepQ can also result in bedaquiline resistance in *M. tuberculosis* [16]. Therefore, the detection of mutations in genes associated with drug resistance can facilitate the implementation of early treatment strategies for tuberculosis, ultimately improving the prognosis of the disease. Whole-genome sequencing is a valuable tool for accurately identifying drug-resistant strains of tuberculosis by predicting drug susceptibility, gene functionality, resistance mechanisms, and other relevant characteristics [17]. Despite the proven efficacy of bedaquiline in treating tuberculosis, there is a lack of comprehensive information on tuberculosis treatment, including the absence of established guidelines for antibiotic concentrations against clinical strains *M. tuberculosis*. The identification of the bedaquiline resistance gene in *M. tuberculosis*, the initial site of its widespread application, establishes a significant milestone in the understanding of the efficacy of bedaquiline and other diarylquinoline-based medications. This study will lay the foundation for the future utilization of bedaquiline in clinical settings.

## Materials and methods

From February 2018 and July 2019, a total of 2,136 cases containing *M. tuberculosis* strains were collected from clinical samples at Lanzhou Pulmonary Hospital (Gansu, China). The *Tuberculosis Diagnosis and Treatment Guidelines* established by the Chinese Medical Association's Tuberculosis Society were utilized as the basis for diagnosis. All samples were systematically coded, with the investigation focusing solely on the demographic variables of sex and age. The yinke medicine supplied the Lowenstein-Jensen medium kit for the cultivation of drug-resistant *M. tuberculosis*, while a liquid culture media for mycobacteria was procured from Beso Biotechnology.

### Drug resistance detection

Selected *M. tuberculosis* colonies in the logarithmic stage of growth were transferred into grinding tubes containing glass beads and 0.5% Tween80, which were then mounted on oscillators and agitated to achieve a cheese-like appearance. The bacteria were subsequently diluted with normal saline and combined with a bacterial suspension of $1\times10^8$CFU/mL. Further dilutions were made to achieve concentrations of $1\times10^6$CFU/mL and $1\times10^4$CFU/mL for

turbidimetric analysis in a McBurnish tube. Ten drugs, such as isoniazid, rifampicin, ethambutol, streptomycin, kanamycin, levofloxacin, capreomycin, amikacin, moxifloxacin, and pyrazinamide, were assessed for drug resistance in all samples through the proportionate method. The bacterial liquid was evenly spread over the medium containing different anti-tuberculosis medications using a single ring (0.01 mL) of the diluted bacterial solution and 22 SWG standard inoculation rings, respectively, following the marking approach. Following injection, the specimens were cultivated at 37˚C, and the results were documented after a period of four weeks. Various criteria for assessing medication resistance include monoresistance (resistance to one first-line anti-tuberculosis drug only), polydrug resistance ((resistance to more than one first-line anti-tuberculosis drug (other than both isoniazid and rifampicin)), multidrug resistance (resistance to at least both isoniazid and rifampicin), and extensive drug resistance (resistance to any fluoroquinolone and to at least one of three second-line injectable drugs (capreomycin, kanamycin and amikacin), in addition to multidrug resistance).

## Drug sensitivity detection

Two hundred eight strains exhibiting known phenotypic resistance were randomly selected for analysis, and their resistance to bedaquiline was assessed in liquid culture media using the microporous plate dilution method. A standard control was established by introducing 100 uL of liquid medium into the second microplate well. Bedaquiline was serially diluted concentrations ranging from 4–0.008 g/mL in 10-fold increments from the third to the twelfth Well, with 100 uL of the diluted solution added to each well for testing. The inoculation of 100 uL of diluted bacterial liquid ($1\times10^6$CFU/mL) into wells two through twelve resulted in a final drug concentration that ranged from 0.004μg/mL to 2μg/mL and a bacterial count of $5\times10^5$CFU/mL. Following inoculation, resazurin was added to each well and the microporous plates were inoculated at 37˚C for one week. Results were then reported within 24 hours. The MIC of the medicine against the strain is believed to be the highest concentration well exhibiting a color change from blue to red.

## DNA extraction and sequencing

A nucleotide was extracted from the chosen strains utilizing the Bacterial Genome DNA Extraction Kit following the guidelines provided by the manufacturer (Tiangen Biotech Co., LTD). The three primary drug-resistant gene sequences were amplified using PCR. The primer sequences for the drug-resistant bedaquiline genes Rv0678, atpE, and pepQ are presented in Table 1. The sequencing of the bedaquiline resistance genes Rv0678, atpE, and pepQ from 208 clinical strains was conducted according to a standardized protocol. The PCR reaction conditions included pre-denaturation at 95˚C for 5 minutes, denaturation at 95˚C for 30 seconds, annealing at 62˚C (Rv0678 and pepQ) and 58˚C (atpE) for 30 seconds, and elongation at 72˚C for 30 seconds, repeated for 35 times. Sanger Sequencing was performed on the PCR products at Lanzhou Tianqi Gene Co., LTD, and the resulting sequences were analyzed using

**Table 1. Primer synthesis of three genes.**

| Primer | Sequence (5´ to 3´) | TM valve(˚C) |
|---|---|---|
| Rv0678_F | GTGAGCGTCAACGACGGG | 62 |
| Rv0678_R | TCAGTCGTCCTCTCCGG | 62 |
| atpE_F | ATGGACCCCACTATCGCTGC | 58 |
| atpE_R | TTACTTGACGGGTGTAGCGAA | 58 |
| PE(F) | GTGACACATTCCCAGCGTCGAG | 62 |
| PE(R) | GGAGCCCGCCAGTAGTGTACC | 62 |

DNAMAM software to align with the *M. tuberculosis* H37Rv (NC 000963.3) reference genome for mutation identification.

The sequencing process utilized Oxford Nanopore Technologies. Specifically, *M. tuberculosis* strains with known resistance phenotypes were cultured to logarithmic growth periods, colonies were selected and stored in freeze tubes using inoculation loops, high-speed centrifugation was employed to ensure colony placement at the bottom of the tube, and the tube were subsequently frozen in liquid nitrogen for a minimum of 3 hours before being sent to Beijing Biomarker Technologies Co., Ltd for sequencing with Oxford Nanopore Technologies. The genome assembly was analyzed using a combination of bioinformatics tools, including RepeatMasker, Prodigal, tRNAscan-SE, GenBlastA, CRTIslandPath-DIMOB, and PhiSpy, to identify a diverse array of repetitive-sequence coding genes, noncoding genes, pseudogenes, and genomic islands. Subsequently, gene sequences were further refined and annotated using the GO, COG, and KEGG databases, as well as proprietary databases such as Cazyme and ARDB.

### Statistical analysis

The experiments were conducted in triplicate, following established protocols for sample collection as referenced in the literature [18,19]. The data from this study were organized in Excel 2017 and analyzed using SPSS 22.0 software. Descriptive analysis of the epidemiological characteristics of tuberculosis patients was performed using Origin 2017 Software.

## Results

### Detection of drug resistance of *M. tuberculosis*

Fig 1 presents statistics derived from 2,136 cases of clinical isolates of *M. tuberculosis*. Analysis revealed that the age range for tuberculosis infection spans from 3 to 88 years, with a peak incidence occurring between the ages of 21 and 30, as depicted in Fig 2. The overall drug resistance rate was 39.04%, with multidrug resistance exhibiting the highest prevalence and ethambutol demonstrating the lowest resistance among the medications studied. As shown in Fig 3, it illustrated the various drug-resistant *M. tuberculosis* such as multidrug resistance (19.99%),

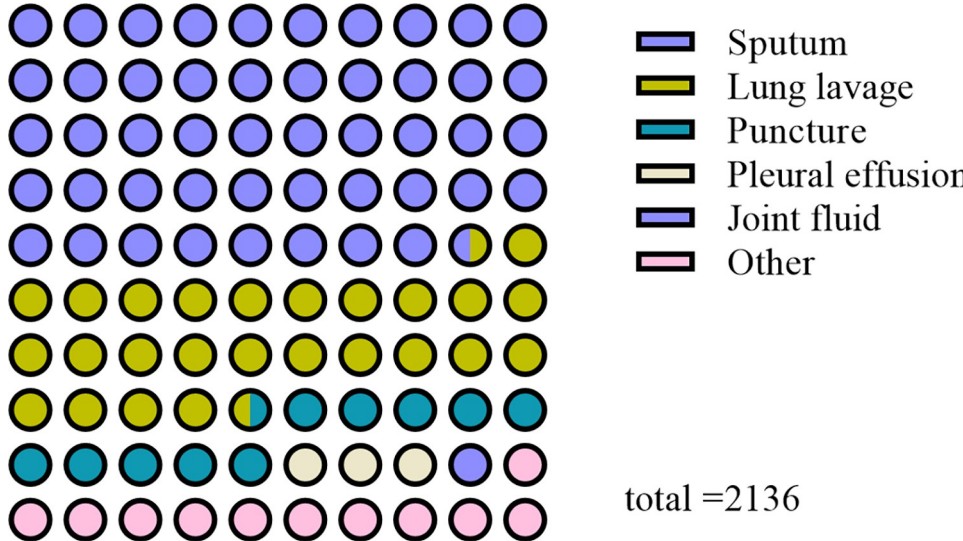

**Fig 1. Distribution of total sample types.**

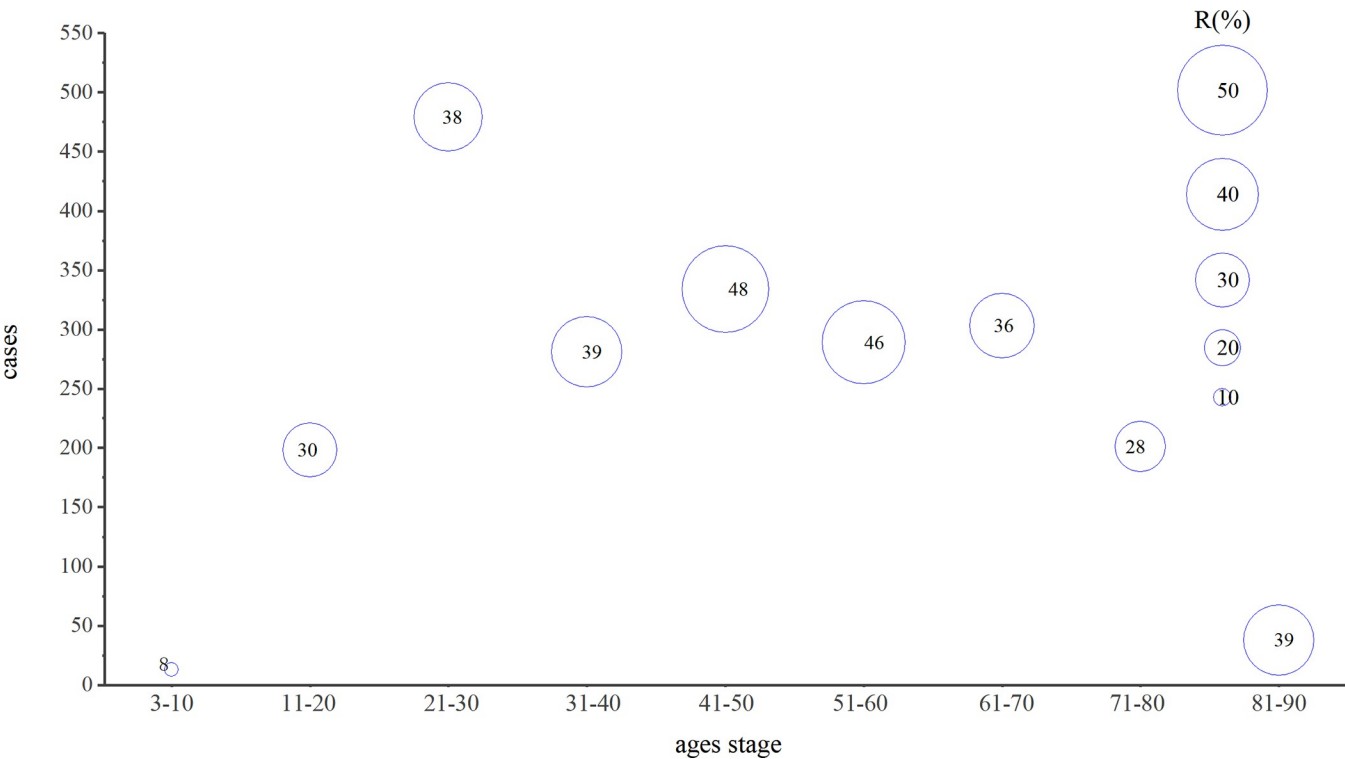

**Fig 2. The distribution of cases at different age stages.** Note: R (%) represents resistance level of samples; the abscissa represents the interval of different ages; the ordinate represents the number of cases at different ages.

polydrug resistance (7.44%), extensive resistance (1.45%), and resistance to streptomycin (4.31%), isoniazid (2.76%), levofloxacin (1.97%), rifampin (0.56%), amikacin or capreomycin (0.37%) and ethambutol (0.19%). By depicting the drug resistance of tuberculosis in different age groups, the results showed that multidrug resistance was higher than other drug resistance types in all age groups, with the exception of the 71–80 stage, with a maximum of 28.74%. Widespread resistance was consistently lower than other types of drug resistance across all age groups, with the highest level observed at 2.99% in the 41–50 age group (Fig 4). Additionally, monoresistance reached a peaked of 13.16% in individuals aged 81 to 90, while resistance to multidrug resistance peaked at 10.45% in the 71–80 age group.

### The complete sequence analysis of *M. tuberculosis*

By conducting an analysis of the entire genome sequence, it was determined that the genes of *M. tuberculosis* were involved in lipid transport and metabolism, amino acid transport and metabolism, as well as energy production and conversion (Figs 5 and 6). Additionally, it was found that the genes responsible for 82.45% of the enzymes related to carbohydrates encode glycoside hydrolases, glycosyl transferases, and carbohydrate esterases, as depicted in Fig 7. Fig 8 illustrated the involvement of ABC transporters, two-component systems, amino acid biosynthesis, carbon metabolism, citric acid cycle, fatty acid biosynthesis, and bacterial secretion systems as key areas of participation. Additionally, this study presented the prediction of *M. tuberculosis* gene map (Fig 9), along with the identification of mutations in the pentapeptide repeat family gene, undecenyl pyrophosphate phosphatase, and aminoglycoside N-acetyltransferase genes of *M. tuberculosis*.

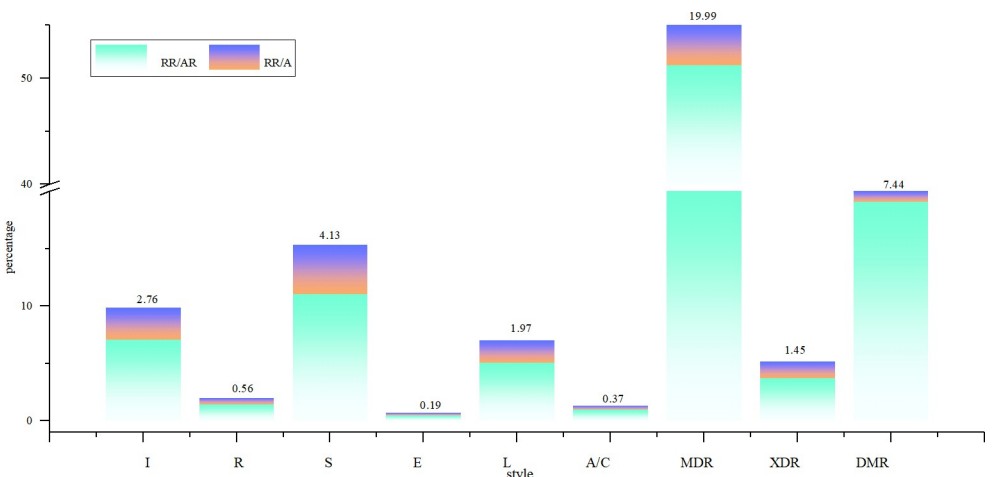

**Fig 3. The distribution of different drug resistance types.** Note: The I, R, S, E, L, A/C, MDR, XDR, DMR are respectively monoresistance to isoniazid, rifampicin, streptomycin, ethambutol, levofloxacin, amikacin or capreomycin, multidrug resistance, extensive resistance, polydrug resistance; The RR, AR and A represent respectively resistance, total resistance and total samples.

## The drug sensitivity tests of bedaquiline to *M. tuberculosis*

A variety of *M. tuberculosis* strains were subjected to testing for bedaquiline resistance using both phenotypic and molecular techniques. The inhibitory concentration of bedaquiline was found to be predominantly at 0.060 g/mL across a range of MIC values, as depicted in Fig 10. Specifically, the MIC of bedaquiline against various clinical strains was determined as follows: 0.481% at 2.000μg/mL, 0.962% at 1.000μg/mL, 3.365% at 0.500μg/mL, 14.423% at 0.250μg/mL,

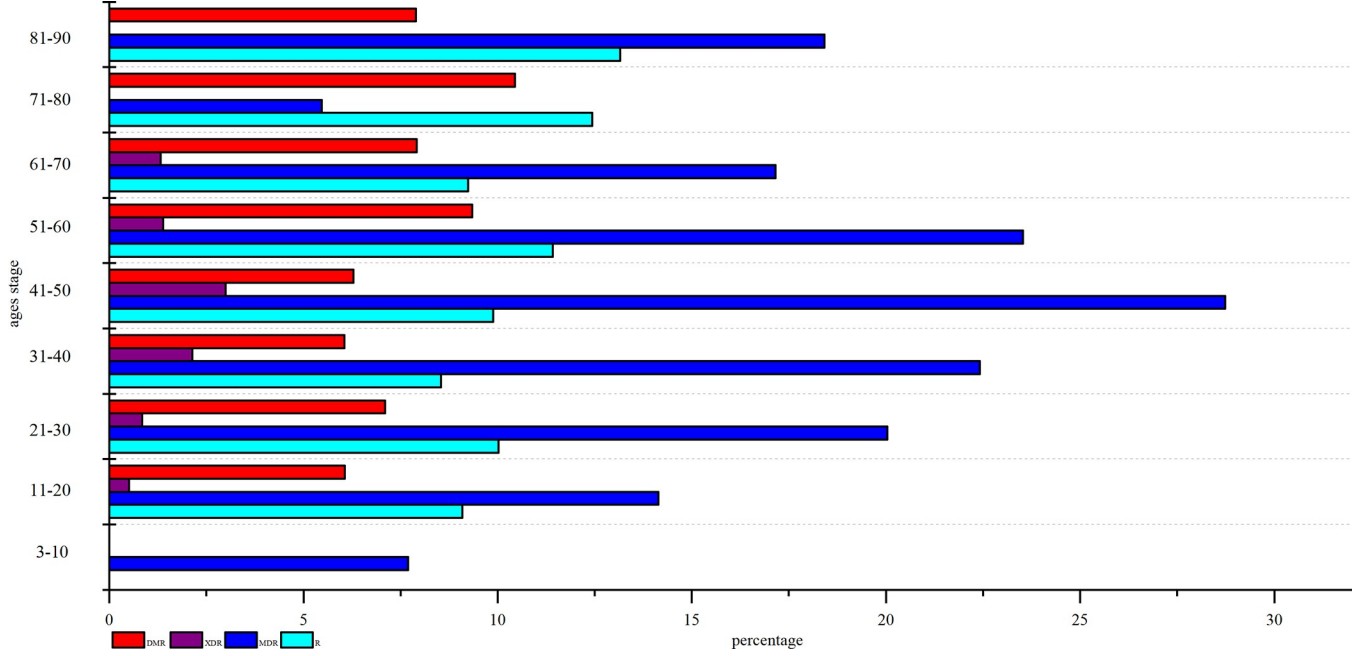

**Fig 4. The distribution of drug resistance types at different age stages.** Note: The DMR, XDR, MDR and R represent respectively polydrug resistance, extensive resistance, multidrug resistance, monoresistance tuberculosis.

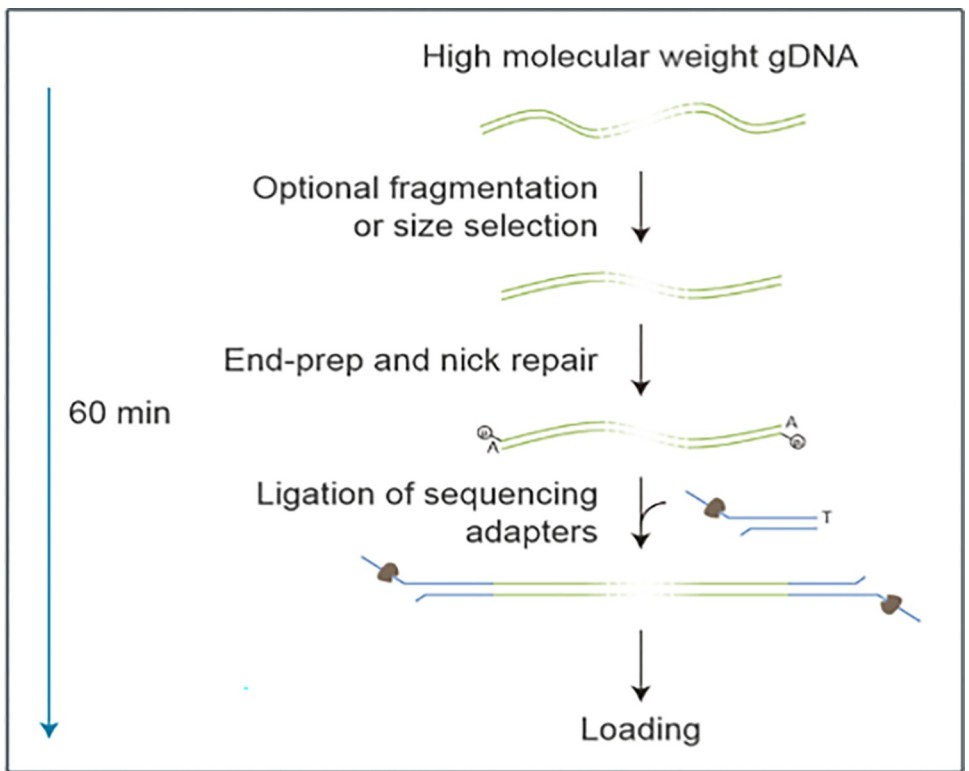

**Fig 5. Experimental sequencing process.**

13.462% at 0.120μg/mL, 50% at 0.060μg/mL, 10.096% at 0.030μg/mL, 1.923% at 0.016μg/mL, 1.442% at 0.008μg/mL, and 3.846% at 0.004μg/mL.

## Bedaquiline drugs resistance gene analysis

In this study, three drug resistance genes were amplified and partial results were presented (Fig 11). The mutation rate was determined as the proportion of test strains exhibiting mutation. The results, as depicted in Table 2 and Fig 12, suggested an association between bedaquiline resistance and Rv0678, atpE, and pepQ. Specifically, the P62A mutation was identified in RV0678 with an 8.3% mutation rate. The atpE gene exhibited a mutation rate of 10.1%, predominantly displaying the V152A mutation, with additional mutations identified as G77R, S51F, E112K, and L141R. Furthermore, analysis of Table 2 revealed that the pepQ gene predominantly harbored the T222N mutation, which had a mutation rate of 4.1%. The highest prevalence of mutations associated with bedaquiline resistance were observed in the AtpE and Rv0678 genes, distributed in a seemingly random pattern throughout the Open Reading Framework.

## Discussion

Tuberculosis poses a significant challenge to global public health, necessitating the implementation of a viable strategy and efficacious treatment measures [20]. To gain a comprehensive understanding of tuberculosis and the resistance *M. tuberculosis* to current anti-tuberculosis medications, as well as to establish a scientific foundation for tuberculosis prevention and treatment, our initial investigation focused on the resistance profiles of established clinical

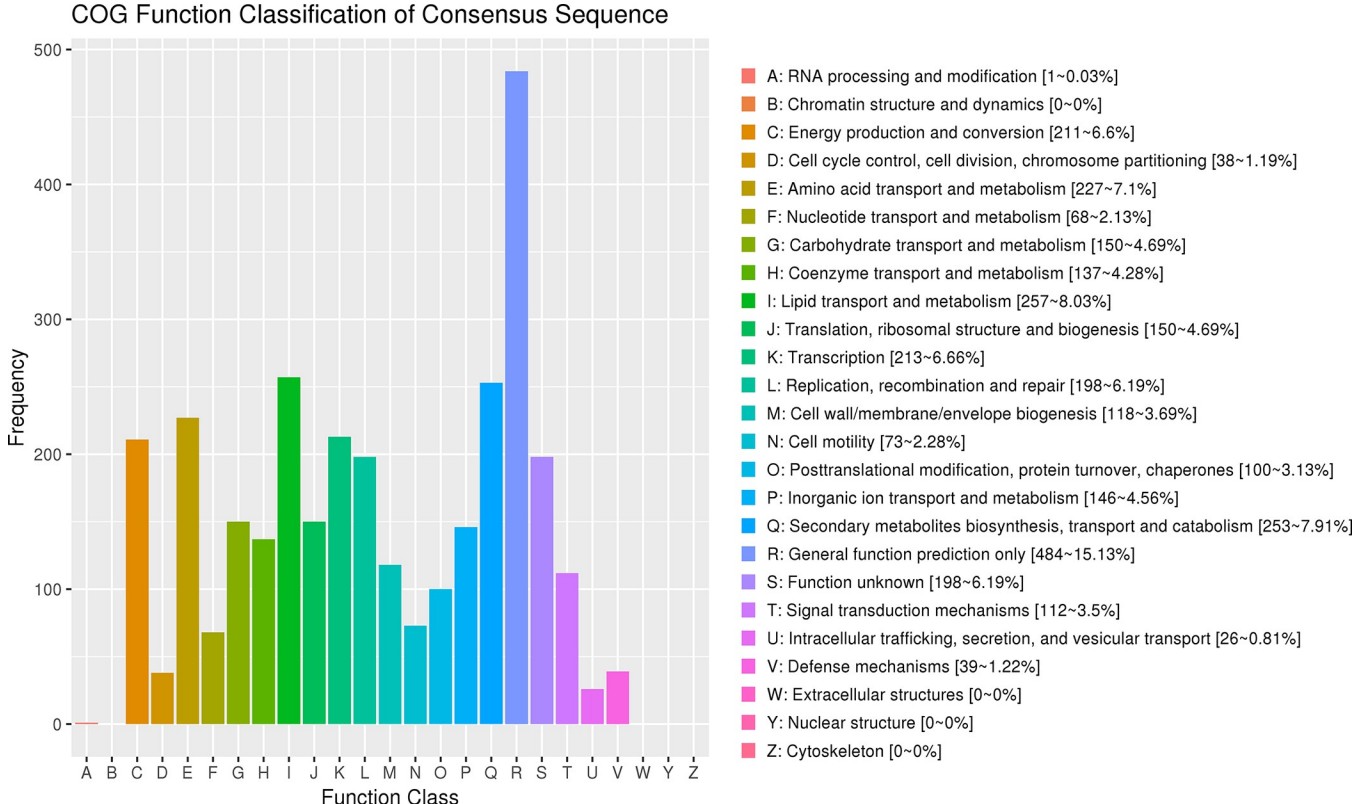

**Fig 6. Protein functional cluster analysis.** Note: The abscissa is the classification content of COG, and the ordinate is the number of genes. In different functional classes, the proportion of genes reflects the metabolic or physiological preferences in the corresponding period and environment.

agents. Subsequently, we examined the inhibitory concentration and drug resistance of bedaquiline against *M. tuberculosis*. The drug resistance model developed from this research may present viable treatment options for tuberculosis and enhance clinical efficacy [21]. The emergence of drug-resistant tuberculosis has posed substantial challenges to the prevention and treatment of tuberculosis. Isoniazid, a key component of first-line tuberculosis therapy, exhibits a notable resistance rate of 2.76% against *M. tuberculosis* [22]. The dynamic alterations and mechanisms associated with isoniazid resistance mutations in *M. tuberculosis* remain incompletely understood, despite the infrequent occurrence of isoniazid mono-resistance. The article indicates a notable increase in the proportion of *M. tuberculosis* strains exhibiting resistance to isoniazid compared to previous years. Furthermore, the escalation in cases of *M. tuberculosis* carrying an inhA mutation resistant to isoniazid may be associated with previous exposure to propioniazid [23,24]. Isoniazid exhibits the highest rate of drug resistance among all substances, with drug resistance rates for tuberculosis relapse surpassing those for tuberculosis onset [25]. In the Xi'an investigation, multidrug resistance accounted for 24.4% of the total resistance of *M. tuberculosis*, which was determined to be 39.0% [26]. Conversely, in Dalian city, there has been a notable reduction in both overall drug resistance and multidrug resistance rates for tuberculosis in recent years, with monoresistance accounting for approximately half of the total drug resistance [27]. The lack of reliability of rifampicin resistance as a biomarker for detecting multidrug resistant tuberculosis necessitates the use of techniques capable of identifying both rifampicin and isoniazid resistance in order to develop an effective treatment plan for tuberculosis [28]. The results indicated an overall tuberculosis drug resistance

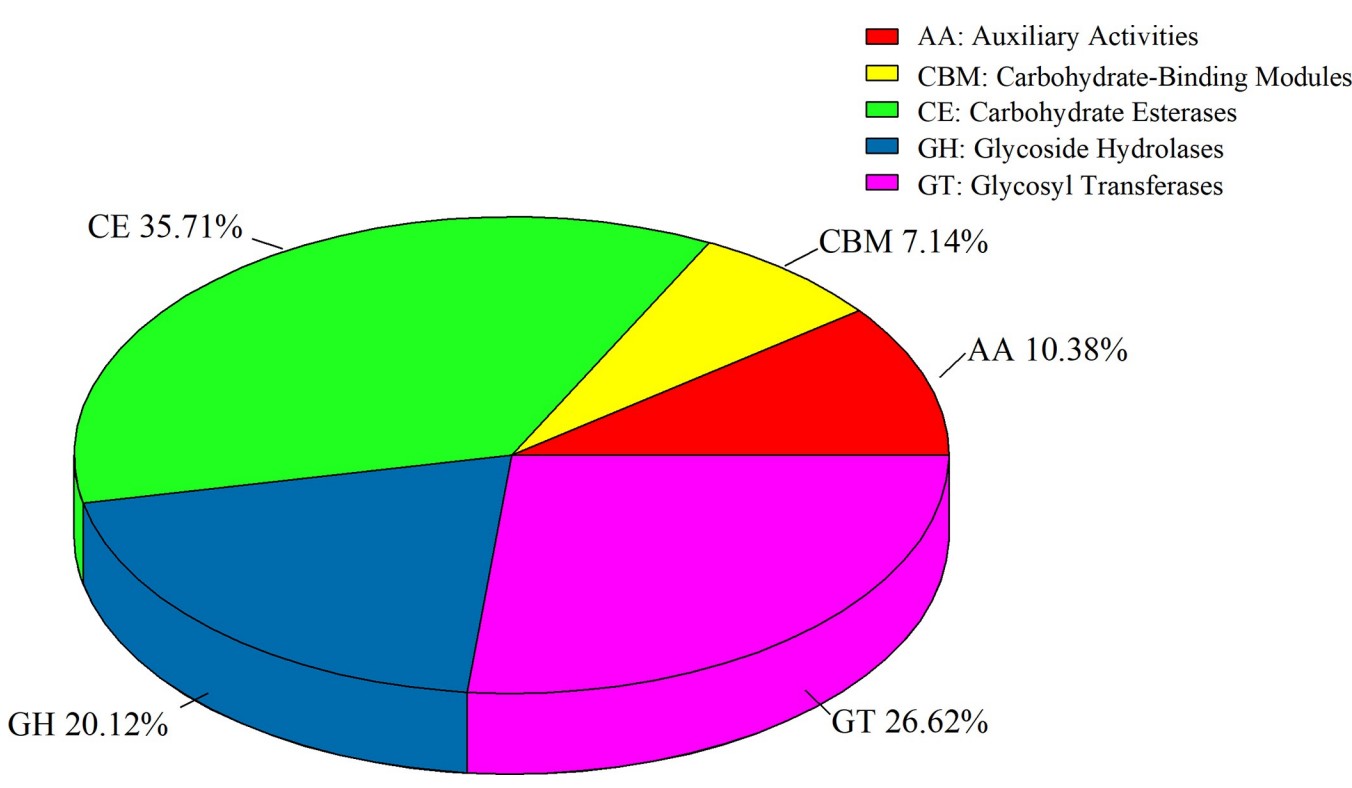

**Fig 7. Carbohydrate-related enzymes classification.**

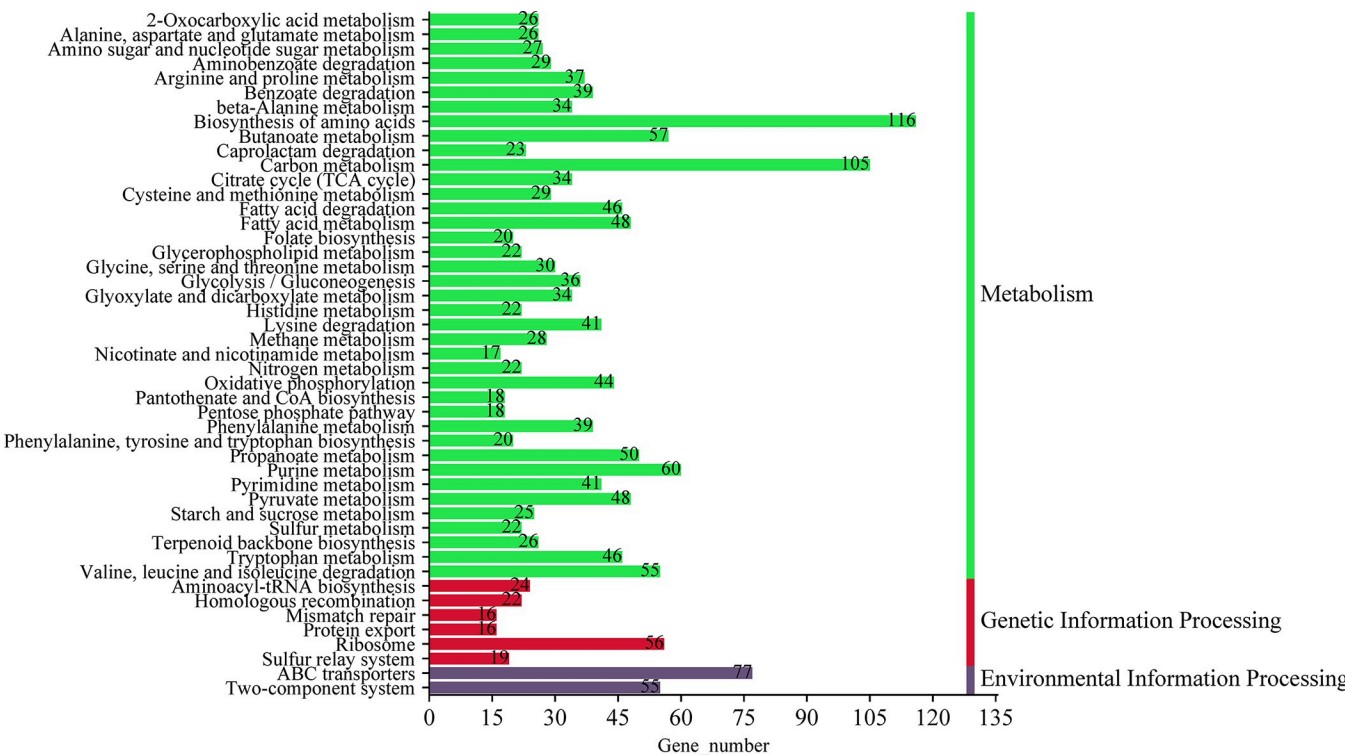

**Fig 8. Analysis of the whole protein KEGG pathway.** Note: The ordinate is KEGG secondary classification and the abscissa is percentage.

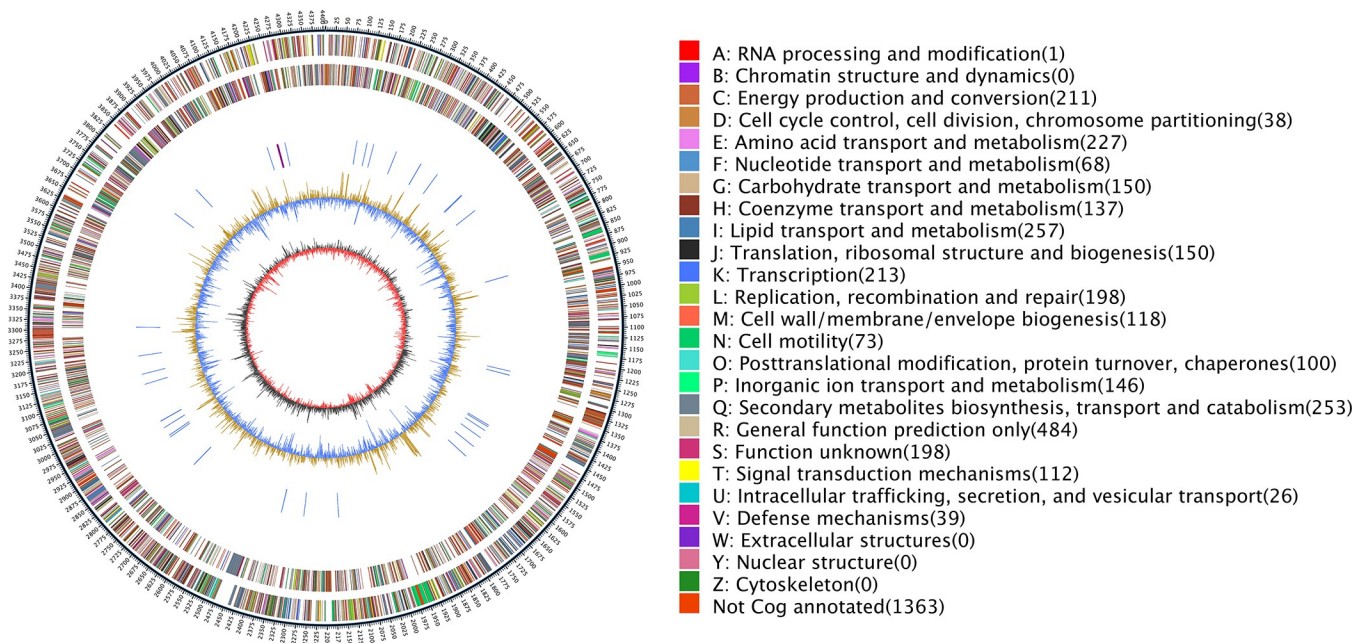

**Fig 9. Genomic cycle map for *M. tuberculosis*.** Note: Circos visualizes the genome to more clearly explore the relationships between components or locations in the genome.

rate of 39.04% and a multidrug resistance rate of 19.99%. Analysis of the samples revealed a higher resistance to ethambutol and streptomycin compared to other monoresistance. The aforementioned findings suggest a potentially elevated rate of tuberculosis drug resistance in

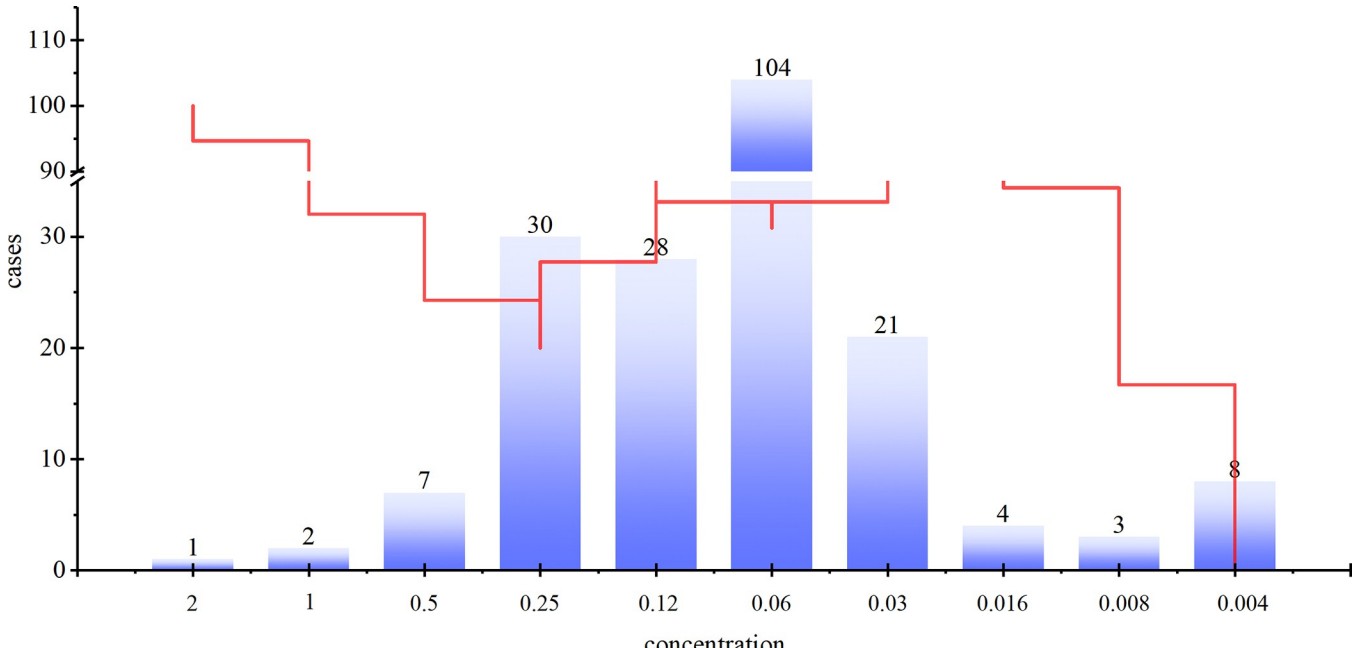

**Fig 10. Antibacterial concentrations of bedaquiline on Mycobacterium tuberculosis.** Note: Red dashed line represents resistance level of samples; The ordinate represents the probability of cases.

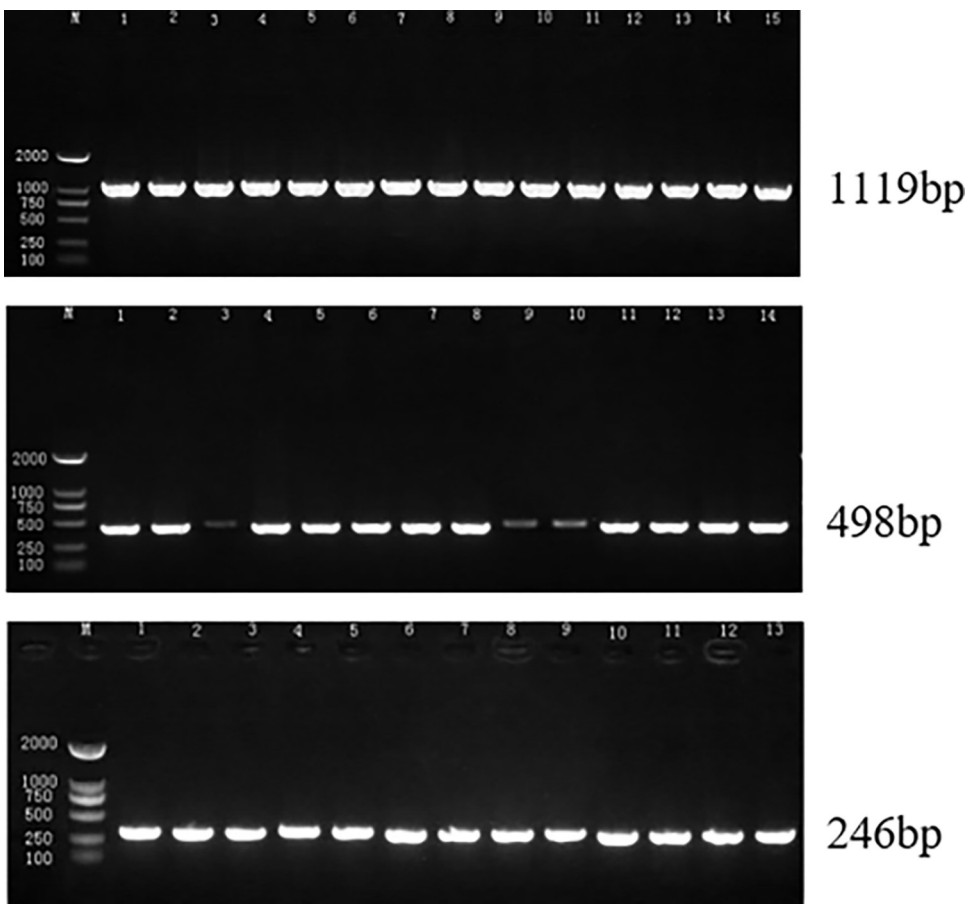

**Fig 11. Agarose gel of three resistant genes.**

the region. The work can offer insight into tuberculosis treatment and establishes the groundwork for *M. tuberculosis* studies in Gansu province.

The utilization of whole-genome sequencing is increasingly essential in the diagnosis and research of tuberculosis due to its ability to rapidly identify medication resistance, characterize strains, and provide data for genotyping. Whole-genome sequencing for diverse *M. tuberculosis* strains demonstrates high accuracy in species identification, consistent mapping of drug resistance compared to drug sensitivity testing, and early detection of drug resistance prior to sensitivity testing [29–31]. The findings of this study indicate that gene mutations are responsible for the bacitracin and fluoroquinolone resistance phenotypes in *M. tuberculosis*. The resistance to these medications, particularly fluoroquinolones, is expected based on theoretical considerations. Further investigation into the mechanisms underlying this resistance is warranted. Whole-genome sequencing can help identify new mutation sites and elucidate the

**Table 2. Mutation rates and mutation sites of drug-resistant genes.**

| gene | Mutation rate | main mutation | other mutation | reported mutations |
|---|---|---|---|---|
| atpE | 8.2 | P62A | D27G, V74L | D28V/P, E61D, I66M |
| Rv0678 | 5.8 | V152A | G77R, S51F, E112K, L141R | S63R/G, R50Q, R107C |
| pepQ | 3.4 | T222N | \ | G265T, T157C, R337L |

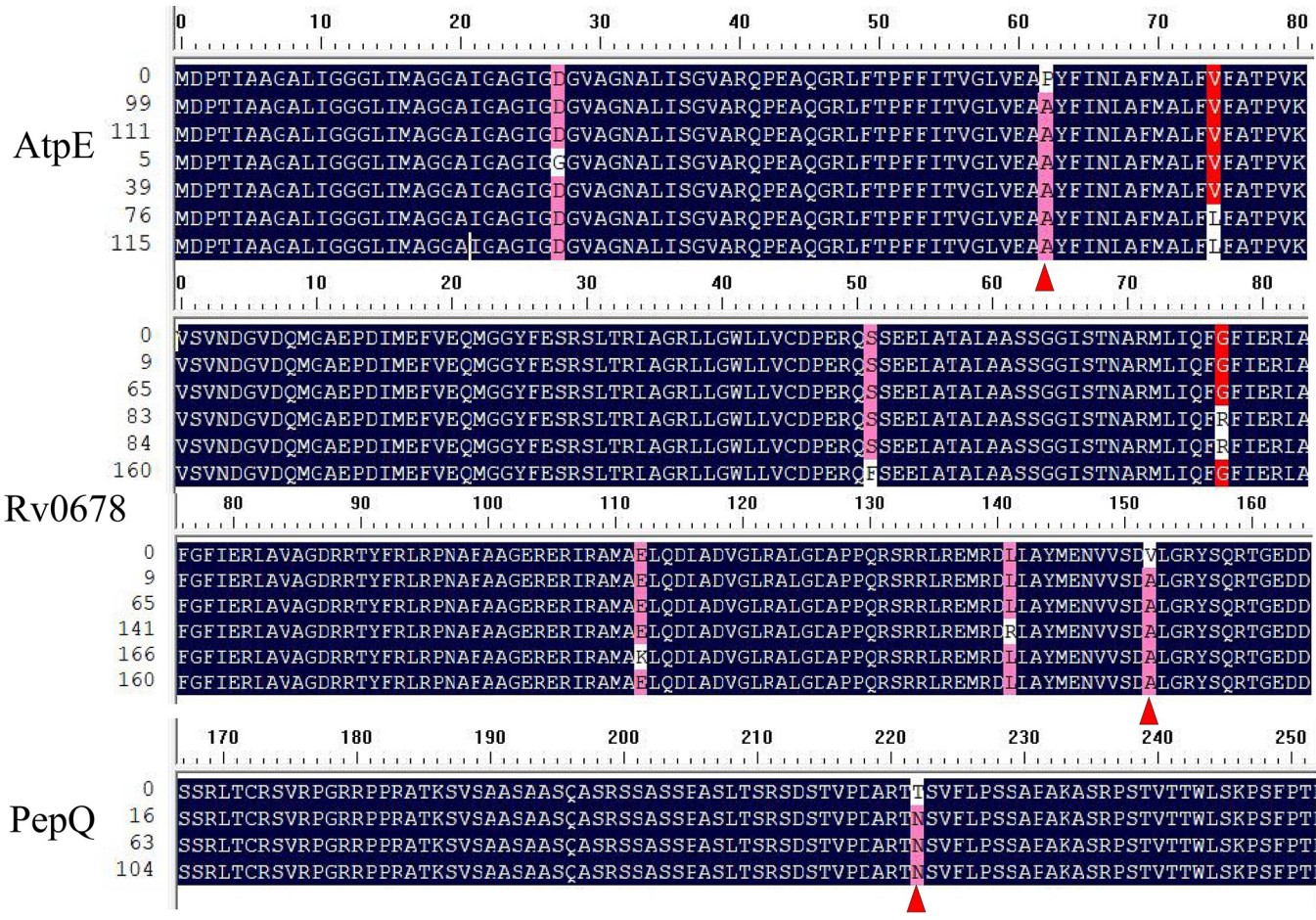

**Fig 12. Alignment the major and other mutation sites of each drug resistance gene.**

pathways leading to drug-resistant mutations, facilitating early detection of drug-resistant tuberculosis. Genome analysis revealed that the ofloxacin resistance in *M. tuberculosis* is not solely attributed to the DNA gyrase mutation, as the V762G mutation also plays a significant role in reducing the drug's sensitivity. Furthermore, the drug efflux mechanism was determined not to be a contributing factor to *M. tuberculosis's* ofloxacin resistance [32]. Whole-genome sequencing can be utilized to identify novel mutations associated with drug resistance. The results demonstrate that the *M. tuberculosis* gene plays a significant role in regulating the primary membrane material transit, center carbon metabolism, and secretion system. The *M. tuberculosis* proprotein transferase subunit family (SecD, SecEThe, SecA, and YajC), membrane insertion protease, fusion signal recognition particle receptor, and signal recognition particle subunit SRP54 are all components of the bacterial secretion system. Analysis of the genome allows for a comprehensive understanding of gene distribution, protein functions, and biological processes within *M. tuberculosis*

The implementation of rapid molecular detection significantly improves the accuracy level of tuberculosis by directly identifying *M. tuberculosis* and drug treatment resistance in clinical specimens. This approach accelerates the administration of anti-tuberculosis drugs and reduces the duration of tuberculosis treatment [33]. Drug sensitivity tests is essential in conjunction with appropriate tuberculosis therapy as it can identify the emergence of bedaquiline

resistance at an early stage [34]. Bedaquiline exhibited potent activity against *M. tuberculosis* with a MIC of 0.062μg/mL, along with moderate efficacy against non-*M. tuberculosis* strains [35]. Resazurin microtitration revealed bedaquiline MIC values ranging from 0.0039 to 0.25 μg/mL against *M. tuberculosis* strains in Latin America [36]. In the present investigation, the MICs of bedaquiline against *M. tuberculosis* exhibited a range of 0.004 μg/mL to 2 μg/mL, with the 50% strain demonstrating a MIC concentration of 0.06 μg/mL, aligning closely with the study's results. Nevertheless, clinical resistance to bedaquiline has been observed in *M. tuberculosis*.The predominant factors contributing to bedaquiline resistance were identified as mutations in either Rv0678 or atpE. Initially, mutations in Rv0678 were associated with low-level resistance, followed by mutations in atpE leading to high-level resistance.

The study results indicated that the Rv0678 and atpE genes exhibited the highest frequency of V152A and P62A mutations, respectively, with the pepQ gene showing the highest occurrence of T222N mutations. The site of mutations in bedaquiline resistance genes detected in this experiment have not been reported in previous studies. It should be noted that these findings are specific to tuberculosis within the national context, as Gansu Province serves as a representative region. Furthermore, alterations in the Rv0678 and atpE genes were identified, and treatment with bedaquiline resulted in an increase in the MICs against *M. tuberculosis* [37]. The *M. tuberculosis* strains exhibiting extensive drug resistance have shown notable efficacy against bedaquiline and clofazmin in China. Early detection and treatment of tuberculosis may contribute to the emergence of bedaquiline resistance [38]. The study results indicate that bedaquiline is an effective treatment for tuberculosis, with a critical resistance concentration of 0.06 μg/mL. These findings offer valuable insights for future research and utilization of bedaquiline in the management of *M. tuberculosis* infections.

## Supporting information

**S1 File.**
(XLSX)

**S1 Raw image.**
(PDF)

## Acknowledgments

We thank Chongxiang Tong, attending physicians at Lanzhou Pulmonary Hospital, for their useful discussions and support. We would also like to thank Bo Yu, President, for providing the experimental platform at the Second People's Hospital of Lanzhou City. We would also like to thank Jin Wu, physicians at Traditional Chinese Medicine Hospital of Lanzhou City.

## Author Contributions

**Conceptualization:** Xiaoning Huo.

**Data curation:** Chenchen Li, Xueke Hui.

**Formal analysis:** Chenchen Li.

**Funding acquisition:** Zhong Jia.

**Supervision:** Xiaoning Huo.

**Writing – original draft:** Yousheng Peng.

**Writing – review & editing:** Nigus Abebe Shumuyed.

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
