## [Decision Letter · Decision Letter 0]

17 Apr 2024

PONE-D-23-36124The drug susceptibility and gene-set enrichment analysis of M. tuberculosis isolates in Gansu Province, ChinaPLOS ONE

Dear Dr. Jia,

Thank you for submitting your manuscript to PLOS ONE. After careful consideration, we feel that it has merit but does not fully meet PLOS ONE’s publication criteria as it currently stands. Therefore, we invite you to submit a revised version of the manuscript that addresses the points raised during the review process.

We look forward to receiving your revised manuscript.

Kind regards,

Salman Sadullah Usmani, Ph.D.

Academic Editor

PLOS ONE

3. PLOS requires an ORCID iD for the corresponding author in Editorial Manager on papers submitted after December 6th, 2016. Please ensure that you have an ORCID iD and that it is validated in Editorial Manager. To do this, go to ‘Update my Information’ (in the upper left-hand corner of the main menu), and click on the Fetch/Validate link next to the ORCID field. This will take you to the ORCID site and allow you to create a new iD or authenticate a pre-existing iD in Editorial Manager. Please see the following video for instructions on linking an ORCID iD to your Editorial Manager account: https://www.youtube.com/watch?v=_xcclfuvtxQ".

4. We suggest you thoroughly copyedit your manuscript for language usage, spelling, and grammar. If you do not know anyone who can help you do this, you may wish to consider employing a professional scientific editing service.

A clean copy of the edited manuscript (uploaded as the new *manuscript* file)”.

6. In the online submission form, you indicated that [The data underlying the results presented in the study are available from author].

7. Ethics statement only appears at the end of the manuscript:

Your ethics statement should only appear in the Methods section of your manuscript. If your ethics statement is written in any section besides the Methods, please move it to the Methods section and delete it from any other section. Please ensure that your ethics statement is included in your manuscript, as the ethics statement entered into the online submission form will not be published alongside your manuscript.

8. Thank you for stating the following financial disclosure:

 [Zhong Jia, Natural Science Foundation of Gansu Province (18JR3RA417)].

9. Thank you for stating the following in the Acknowledgments Section of your manuscript:

[The study was supported by Natural Science Foundation of Gansu Province (18JR3RA417).]

 [Zhong Jia, Natural Science Foundation of Gansu Province (18JR3RA417)].

Additional Editor Comments:

The manuscript has been reviewed by four reviewers, who identified significant concerns in the study's design and methodology. Mixing pulmonary and extrapulmonary isolates for analysis is problematic, as bedaquiline is intended for treating pulmonary multidrug-resistant tuberculosis (MDR-TB) according to CDC guidelines. This approach may lead to unreliable conclusions. The study also lacks comparison with existing research on mutations in the Rv0678, atpE, and pepQ genes, limiting the novelty and validity of the findings. Additionally, there is clear issues with the writing quality and data presentation, including unclear language, grammatical errors, and improper formatting of scientific terms. Some figures lack sufficient detail or resolution, making interpretation challenging.

Reviewers' comments:

Reviewer's Responses to Questions

**Comments to the Author**

1. Is the manuscript technically sound, and do the data support the conclusions?

Reviewer #1: Partly

Reviewer #2: Yes

Reviewer #3: No

Reviewer #4: Partly

2. Has the statistical analysis been performed appropriately and rigorously?  Reviewer #1: N/A

Reviewer #2: Yes

Reviewer #3: I Don't Know

Reviewer #4: No

3. Have the authors made all data underlying the findings in their manuscript fully available?

Reviewer #1: No

Reviewer #2: Yes

Reviewer #3: No

Reviewer #4: No

4. Is the manuscript presented in an intelligible fashion and written in standard English?

Reviewer #1: No

Reviewer #2: Yes

Reviewer #3: No

Reviewer #4: No

5. Review Comments to the Author

Reviewer #1: The drug susceptibility and gene-set enrichment analysis of M. tuberculosis isolates in

Gansu Province, China

The article seem to have a very good data that is not well presented starting from the title through the rest of the content. The abstract does not tell us clearly what the authors did, why, how and what did they conclude.

The methods regarding the bioinformatics section is very abstract and does not tell us how and why these list of databases and software are used.

Reviewer #2: Dear authors, as much I can evaluate your manuscript it can be accepted. However, at some of the area of your manuscript, I do not have expertise which needs to be evaluated by other reviewers.

Kindly consider the other reviewer comments seriously.

Reviewer #3: The authors describe that Bedaquiline is a new drug in the paper and carried out PCR and whole genome sequencing of isolates. This paper has multiple issues.

1. The authors are using pulmonary and extra pulmonary isolates together to interpret the data which does not make sense as the site of infection are different.

2. Bedaquiline has been found to be useful drug since 2016 not sure why the authors say it is a new drug.

3. The authors dont show any comparison with other studies for the mutations they found in Rv0678, atpE, and pepQ genes as there are multiple studies which have reported mutation in these genes from several isolates from different countries.

4. In MTB most of the genes show enrichment of of ABC transporters, two-component systems, and bacterial

secretion systems what new information are the authors adding in this study.

5. Bedaquiline may be used to treat adults with a confirmed diagnosis of pulmonary MDR TB as per the CDC guideline so why did the authors use extra pulmonary isolates in the study.

6. There are several grammatical errors in the paper for example M. tuberculosis is not italicized in the entire paper.

Reviewer #4: The study aimed to investigate various aspects of tuberculosis, including drug resistance, sensitivity, and genetic mutations, using samples collected from Lanzhou Pulmonary Hospital in Gansu, China, between February 2018 and July 2019. A total of 2,136 cases with M. tuberculosis strains were analyzed, following the Tuberculosis Diagnosis and Treatment Guidelines of the Chinese Medical Association.

Drug resistance detection was conducted using a proportionate approach for ten drugs, and drug sensitivity testing for bedaquiline was performed on 208 strains. DNA extraction and sequencing were carried out to identify drug-resistant gene sequences, with subsequent analysis to predict gene function and resistance mechanisms. Results indicated varying degrees of drug resistance, including multi-drug resistance and extensive resistance, among the samples. Mutations in specific genes, such as Rv0678, atpE, and PepQ, were associated with bedaquiline resistance. Additionally, pathway analysis revealed potential mechanisms underlying drug resistance.

1. Please use help of English speaker or any grammatical tools to improve the quality of the manuscript. Below are some examples which should be implemented to improve the quality. "Demonstrated a polymorphic distribution with different regions" - It's not clear what is meant by "polymorphic distribution. The phrase "were firstly predicted" should be "were predicted initially" or "were predicted first."

2. "Bedaquiline's, as a new drug, resistance range was identified" - This sentence is unclear. It could be rephrased as "The range of resistance to bedaquiline, a new drug, was identified."

3. Please elaborate “82.45% of the enzymes involved in carbohydrates”.

4. Write more clearly “the effects of drug-resistant tuberculosis cases” at Page 5 line 5?

5. Some images are not clear such as Figure 9 and 12

6. Figure 8 is not full please provide full image for pathway analysis.

7. “Note: A, B, C and D are amino acid comparison diagrams of atpE, Rv0678 and PepQ genes, respectively. 0 represents the amino acid sequence of the corresponding drug-resistant gene of the standard strain of M. tuberculosis. The other numbers represent the coding of the corresponding strain.” if it belongs to Figure 12 then modify it accordingly.

8. Data analysis methodology can be incorporated in more details for better understanding

9. The description of drug resistance categories (single drug resistance, multi-drug resistance, etc.) lacks clarity and standardization.

6. PLOS authors have the option to publish the peer review history of their article (what does this mean?). If published, this will include your full peer review and any attached files.

Reviewer #1: **Yes: **Rehab Ahmed

Reviewer #2: **Yes: **Shaban Ahmad

Reviewer #3: No

Reviewer #4: No

---

## [Author Response · Author response to Decision Letter 0]

12 Jun 2024

Dear Editors and Reviewers:

Thank you for your comments and suggesting concerning our manuscript entitled “The drug susceptibility and gene-set enrichment analysis of M. tuberculosis isolates in Gansu, China” (PONE-D-23-36124R1). Those comments are all valuable and very helpful for revising and improving our paper, as well as the important guiding significance to our researches. We have responded carefully to all comments by the reviewers and have made correction which we hope meet with approval. Revised portion are marked in red in the paper.

Response to comments from the Editor

Additional Editor Comments:

The manuscript has been reviewed by four reviewers, who identified significant concerns in the study's design and methodology. Mixing pulmonary and extrapulmonary isolates for analysis is problematic, as bedaquiline is intended for treating pulmonary multidrug-resistant tuberculosis (MDR-TB) according to CDC guidelines. This approach may lead to unreliable conclusions. The study also lacks comparison with existing research on mutations in the Rv0678, atpE, and pepQ genes, limiting the novelty and validity of the findings. Additionally, there is clear issues with the writing quality and data presentation, including unclear language, grammatical errors, and improper formatting of scientific terms. Some figures lack sufficient detail or resolution, making interpretation challenging.

Response: We thank the editor for their constructive comments. The experimental M. tuberculosis samples utilized in this study were sourced from various biological sources, including sputum, lung lavage, puncture, pleural effusion, and joint fluid, obtained from in vitro culture methods. The reason that the strains used in the bedaquiline susceptibility experiments do not only include multidrug-resistant strains was to investigate the effectiveness of bedaquiline resistance against M. tuberculosis. The sites of bedaquiline resistance gene mutations detected in this experiment have not been reported in previous studies and belong to novel gene mutations, which was modified in the discussion section. In addition, we have asked a colleague whose native language is English to review our manuscript, revising the inappropriate language and format.

Response to comments from the Reviewer

Reviewer #1: The drug susceptibility and gene-set enrichment analysis of M. tuberculosis isolates in

Gansu Province, China

The article seem to have a very good data that is not well presented starting from the title through the rest of the content.

Response: Thanks for the helpful comments. We reviewed the article again and changed the title to “Phenotypic and genotypic analysis of drug resistance in M. tuberculosis isolates in Gansu, China”. We hope that this title summarized the subject of the article.

The abstract does not tell us clearly what the authors did, why, how and what did they conclude.

Response: Thanks for the suggestion. We modified the summary, and this experiment first investigated the drug resistance in tuberculosis, then tested the antibacterial effect of bedaquiline on M. tuberculosis, and finally used genome sequencing to predict potential resistance sites. It was concluded that tuberculosis had presented a high level of drug resistance. Phenotypic resistance was related to genes, existing potential genetic resistance in M. tuberculosis. Bedaquiline was found to possess effective antibacterial properties against M. tuberculosis.

The methods regarding the bioinformatics section is very abstract and does not tell us how and why these list of databases and software are used.

Response: Thanks for the great suggestion. We used whole-genome sequencing technology to search for M. tuberculosis gene function and potential drug resistance loci through genome annotation and enrichment.

Reviewer #2: Dear authors, as much I can evaluate your manuscript it can be accepted. However, at some of the area of your manuscript, I do not have expertise which needs to be evaluated by other reviewers.

Kindly consider the other reviewer comments seriously.

Response: Thanks for reviewing our manuscript. We must adopt the suggestions of the reviewer to carefully revise the manuscript carefully.

Reviewer #3: The authors describe that Bedaquiline is a new drug in the paper and carried out PCR and whole genome sequencing of isolates. This paper has multiple issues.

1. The authors are using pulmonary and extra pulmonary isolates together to interpret the data which does not make sense as the site of infection are different.

Response: We thank review’s comment. The experimental M. tuberculosis samples utilized in this study were sourced from various biological sources, including sputum, lung lavage, puncture, pleural effusion, and joint fluid, obtained from in vitro culture methods. Our interpretation was that the source of clinical M. tuberculosis had not only the lungs but also other.

2. Bedaquiline has been found to be useful drug since 2016 not sure why the authors say it is a new drug.

Response: We are grateful for this review’s comment. We were very sorry to have made such a mistake at the beginning and have now revised it in the article.

3. The authors dont show any comparison with other studies for the mutations they found in Rv0678, atpE, and pepQ genes as there are multiple studies which have reported mutation in these genes from several isolates from different countries.

Response: Thank this review for the critical comment. The study results indicated that the Rv0678 and atpE genes exhibited the highest frequency of V152A and P62A mutations, respectively, with the pepQ gene showing the highest occurrence of H211Q mutations. The site of mutations in bedaquiline resistance genes detected in this experiment have not been reported in previous studies. We have made the changes in the text.

4. In MTB most of the genes show enrichment of of ABC transporters, two-component systems, and bacterial secretion systems what new information are the authors adding in this study.

Response: We are thankful for this comment. Our primary aim was to predict potential resistance sites in M. tuberculosis.

5. Bedaquiline may be used to treat adults with a confirmed diagnosis of pulmonary MDR TB as per the CDC guideline so why did the authors use extra pulmonary isolates in the study.

Response: Thank this review for the critical comment. The strains used in the experiments were all derived from in vitro cultures of different types of samples. All the subjects tested are M. tuberculosis.

6. There are several grammatical errors in the paper for example M. tuberculosis is not italicized in the entire paper.

Response: We appreciate these comments. We have changed the M. tuberculosis to italics in the text.

Reviewer #4: The study aimed to investigate various aspects of tuberculosis, including drug resistance, sensitivity, and genetic mutations, using samples collected from Lanzhou Pulmonary Hospital in Gansu, China, between February 2018 and July 2019. A total of 2,136 cases with M. tuberculosis strains were analyzed, following the Tuberculosis Diagnosis and Treatment Guidelines of the Chinese Medical Association.

Drug resistance detection was conducted using a proportionate approach for ten drugs, and drug sensitivity testing for bedaquiline was performed on 208 strains. DNA extraction and sequencing were carried out to identify drug-resistant gene sequences, with subsequent analysis to predict gene function and resistance mechanisms. Results indicated varying degrees of drug resistance, including multi-drug resistance and extensive resistance, among the samples. Mutations in specific genes, such as Rv0678, atpE, and PepQ, were associated with bedaquiline resistance. Additionally, pathway analysis revealed potential mechanisms underlying drug resistance.

1. Please use help of English speaker or any grammatical tools to improve the quality of the manuscript. Below are some examples which should be implemented to improve the quality. "Demonstrated a polymorphic distribution with different regions" - It's not clear what is meant by "polymorphic distribution. The phrase "were firstly predicted" should be "were predicted initially" or "were predicted first."

Response: Thank you for your very helpful suggestions to improve our paper. We have revised this paragraph. We have modified the “firstly prediction” to “predicted” in the text.

2. "Bedaquiline's, as a new drug, resistance range was identified" - This sentence is unclear. It could be rephrased as "The range of resistance to bedaquiline, a new drug, was identified."

Response: Thank the review for the comment. We made changes in the text to rewrite “bedaquiline’s drug resistance range” to “the drug resistance range of bedaquiline”.

3. Please elaborate “82.45% of the enzymes involved in carbohydrates”.

Response: We thank the review for this comment. In this experiment, carbohydrates-related enzymes of M. tuberculosis genes encoding 82.45% were enriched by whole genome sequencing, such as glycoside hydrolases, glycosyl transferases, and carbohydrate esterases.

4. Write more clearly “the effects of drug-resistant tuberculosis cases” at Page 5 line 5?

Response: We appreciate this comment. We feel sorry to make this mistake in the old version. We have made the changes in the text.

5. Some images are not clear such as Figure 9 and 12

Response: Thank reviewer for the comment. We uploaded the high-resolution Figure 9 and Figure 12 when submitting the revision.

6. Figure 8 is not full please provide full image for pathway analysis.

Response: Thanks for the valuable comments. We uploaded the complete figure 8 of the data when we submitted the revised manuscript.

7. “Note: A, B, C and D are amino acid comparison diagrams of atpE, Rv0678 and PepQ genes, respectively. 0 represents the amino acid sequence of the corresponding drug-resistant gene of the standard strain of M. tuberculosis. The other numbers represent the coding of the corresponding strain.” if it belongs to Figure 12 then modify it accordingly.

Response: Thanks again for the valuable comments and helpful suggestions. We have removed the inappropriate paragraphs.

8. Data analysis methodology can be incorporated in more details for better understanding

Response: We are grateful for this review’s comment. A retrospective analysis of the tuberculosis clinical database from February 2018 to July 2019 used tuberculosis diagnosis and treatment guidelines as inclusion criteria.

9. The description of drug resistance categories (single drug resistance, multi-drug resistance, etc.) lacks clarity and standardization.

Response: We are thankful for this comment. We defined monoresistance, polydrug resistance, multidrug resistance, and extensive drug resistance with rereference to “WHO. Definitions and reporting framework for tuberculosis [M].2013.WHO, 2014 : 5”.

We believe that these revisions have significantly improved the quality and clarity of the manuscript. However, if you have any further suggestions or require additional clarification, please do not hesitate to let us know. We are committed to making the necessary modifications to ensure that the manuscript meets the standards of PLOS ONE.

Thank you again for your valuable feedback and support.

Best regards,

Prof. Zhong Jia

College of Veterinary Medicine

Gansu Agricultural University

Lanzhou 730070

PR China

2024.06.12

---

## [Decision Letter · Decision Letter 1]

19 Jul 2024

PONE-D-23-36124R1Phenotypic and genotypic analysis of drug resistance in M. tuberculosis isolates in Gansu, ChinaPLOS ONE

Dear Dr. Jia,

Thank you for submitting your manuscript to PLOS ONE. After careful consideration, we feel that it has merit but does not fully meet PLOS ONE’s publication criteria as it currently stands. Therefore, we invite you to submit a revised version of the manuscript that addresses the points raised during the review process.

We look forward to receiving your revised manuscript.

Kind regards,

Salman Sadullah Usmani, Ph.D.

Academic Editor

PLOS ONE

Additional Editor Comments:

There is still concern raised by one of the reviwer about the combined analysis of pulmonary and extra pulmonary isolates. It will be wise to show these analysis seperately, as the site of infections are different.

Reviewers' comments:

Reviewer's Responses to Questions

**Comments to the Author**

1. If the authors have adequately addressed your comments raised in a previous round of review and you feel that this manuscript is now acceptable for publication, you may indicate that here to bypass the “Comments to the Author” section, enter your conflict of interest statement in the “Confidential to Editor” section, and submit your "Accept" recommendation.

Reviewer #3: (No Response)

Reviewer #4: All comments have been addressed

2. Is the manuscript technically sound, and do the data support the conclusions?

Reviewer #3: No

Reviewer #4: Yes

3. Has the statistical analysis been performed appropriately and rigorously?  Reviewer #3: N/A

Reviewer #4: I Don't Know

4. Have the authors made all data underlying the findings in their manuscript fully available?

Reviewer #3: Yes

Reviewer #4: Yes

5. Is the manuscript presented in an intelligible fashion and written in standard English?

Reviewer #3: Yes

Reviewer #4: Yes

6. Review Comments to the Author

Reviewer #3: 1. The author need to separately analyze pulmonary and extra pulmonary isolates as the site of infections are different and they cannot be mixed up and shown as they all are drug resistant. Show the results separately for pulmonary and extra-pulmonary isolates.

2. The authors need to show comparison of Bedaquiline mutations identified in their study with other WGS studies as Bedaquiline mutations have been also identified in other WGS studies. Please add this information is table2 showing what is unique and common mutations as compared to other studies

Reviewer #4: (No Response)

7. PLOS authors have the option to publish the peer review history of their article (what does this mean?). If published, this will include your full peer review and any attached files.

Reviewer #3: No

Reviewer #4: No

---

## [Author Response · Author response to Decision Letter 1]

1 Aug 2024

Responses to the editor and reviewers comments

Editor (Comments for the Author):

Thank you for submitting your manuscript to PLOS ONE. After careful consideration, we feel that it has merit but does not fully meet PLOS ONE’s publication criteria as it currently stands. Therefore, we invite you to submit a revised version of the manuscript that addresses the points raised during the review process. Please submit your revised manuscript by Sep 02 2024 11:59PM.

Response: Thank you for your consideration and work. The review’s comments are all valuable and very helpful for revising and improving our paper. We have revised the manuscript carefully and responded to the reviewers’ comments point-by-point.

Reviewer #3 (Comments for the Author):

Q1. The author need to separately analyze pulmonary and extra pulmonary isolates as the site of infections are different and they cannot be mixed up and shown as they all are drug resistant. Show the results separately for pulmonary and extra-pulmonary isolates.

Response: Many thanks to the reviewer for the comments. In the diagnosis of tuberculosis, sputum is typically utilized as the primary sample. This is due to the propensity of tuberculosis patients to expel the pathogen from pulmonary lesions into the sputum through coughing and other mechanisms. Consequently, the presence of Mycobacterium tuberculosis and other pertinent evidence can be more directly identified through sputum analysis. However, not all manifestations of tuberculosis are confined to the lungs. Tuberculosis can affect various parts of the body, including the meninges (tuberculous meningitis), bones (bone tuberculosis), and intestines (intestinal tuberculosis). Consequently, diagnostic samples from the affected sites, such as cerebrospinal fluid, bone tissue, and intestinal tissue, can provide a more accurate assessment of the disease's extent and severity. Therefore, reliance solely on sputum samples is insufficient for a comprehensive diagnosis. Moreover, alternative sample types should be employed for patients who are unable to produce sputum or experience difficulty in doing so, to facilitate accurate diagnosis. Utilizing diverse sample types can offer a multifaceted basis for tuberculosis diagnosis. The etiological agent of tuberculosis, irrespective of the infection site, is typically Mycobacterium tuberculosis. Bedaquiline is an antimicrobial agent specifically targeting Mycobacterium tuberculosis. We did not conduct separate analyses of the samples from the lungs and other anatomical regions, as our experimental focus was on Mycobacterium tuberculosis, which exhibits consistent characteristics across the various sample sources. We sincerely hope that this response will satisfy the reviewers.

Q2. The authors need to show comparison of Bedaquiline mutations identified in their study with other WGS studies as Bedaquiline mutations have been also identified in other WGS studies. Please add this information is table2 showing what is unique and common mutations as compared to other studies.

Response: Many thanks to the reviewer for the comments. We reviewed the literature and added in Table 2 the mutation sites of previously identified Bedaquiline resistance genes. We also demonstrate that the mutation sites of the Bedaquiline resistance gene identified in this paper are inconsistent with previously identified mutations (page 9, lines 246-249).

---

## [Decision Letter · Decision Letter 2]

11 Sep 2024

Phenotypic and genotypic analysis of drug resistance in M. tuberculosis isolates in Gansu, China

PONE-D-23-36124R2

Dear Dr. Jia,

We’re pleased to inform you that your manuscript has been judged scientifically suitable for publication and will be formally accepted for publication once it meets all outstanding technical requirements.

Kind regards,

Salman Sadullah Usmani, Ph.D.

Academic Editor

PLOS ONE

Additional Editor Comments (optional):

It seems that the authors have adequately responded to the third reviewer's query and provided a reasonable justification for their analysis. Despite this, the reviewer appears reluctant to accept the paper. However, it is worth noting that the other reviewers are satisfied with the updated version of the manuscript. After two rounds of revisions, I believe the manuscript is now satisfactory and suitable for publication.

Reviewers' comments:

Reviewer's Responses to Questions

**Comments to the Author**

1. If the authors have adequately addressed your comments raised in a previous round of review and you feel that this manuscript is now acceptable for publication, you may indicate that here to bypass the “Comments to the Author” section, enter your conflict of interest statement in the “Confidential to Editor” section, and submit your "Accept" recommendation.

Reviewer #3: (No Response)

Reviewer #4: All comments have been addressed

2. Is the manuscript technically sound, and do the data support the conclusions?

Reviewer #3: No

Reviewer #4: Partly

3. Has the statistical analysis been performed appropriately and rigorously?  Reviewer #3: N/A

Reviewer #4: N/A

4. Have the authors made all data underlying the findings in their manuscript fully available?

Reviewer #3: Yes

Reviewer #4: Yes

5. Is the manuscript presented in an intelligible fashion and written in standard English?

Reviewer #3: Yes

Reviewer #4: Yes

6. Review Comments to the Author

Reviewer #3: The author's have not cited the papers in Table 2 as they have reported the mutations in Bedaquiline.

The extraplumonary and pulmonary samples cannot be combined in the analysis as the author's themselves have written in the response the site of infection is different in pulmonary tuberculosis and in extrapulmonary tuberculosis.

Reviewer #4: (No Response)

7. PLOS authors have the option to publish the peer review history of their article (what does this mean?). If published, this will include your full peer review and any attached files.

Reviewer #3: No

Reviewer #4: **Yes: **Vinod Kumar

---

## [Editor Report · Acceptance letter]

18 Sep 2024

PONE-D-23-36124R2

PLOS ONE

Dear Dr. Jia,

I'm pleased to inform you that your manuscript has been deemed suitable for publication in PLOS ONE. Congratulations! Your manuscript is now being handed over to our production team.

Kind regards,

on behalf of

Dr. Salman Sadullah Usmani

Academic Editor

PLOS ONE